# Tandem riboswitches form a natural Boolean logic gate to control purine metabolism in bacteria

**Madeline E Sherlock[1], Narasimhan Sudarsan[2], Shira Stav[3], Ronald R Breaker[1,2,3]***

[1]Department of Molecular Biophysics and Biochemistry, Yale University, New Haven, United States; [2]Howard Hughes Medical Institute, New Haven, United States; [3]Molecular, Cellular and Developmental Biology, Yale University, New Haven, United States

**Abstract** Gene control systems sometimes interpret multiple signals to set the expression levels of the genes they regulate. In rare instances, ligand-binding riboswitch aptamers form tandem arrangements to approximate the function of specific two-input Boolean logic gates. Here, we report the discovery of riboswitch aptamers for phosphoribosyl pyrophosphate (PRPP) that naturally exist either in singlet arrangements, or occur in tandem with guanine aptamers. Tandem guanine-PRPP aptamers can bind the target ligands, either independently or in combination, to approximate the function expected for an IMPLY Boolean logic gate to regulate transcription of messenger RNAs for de novo purine biosynthesis in bacteria. The existence of sophisticated all-RNA regulatory systems that sense two ancient ribonucleotide derivatives to control synthesis of RNA molecules supports the hypothesis that RNA World organisms could have managed a complex metabolic state without the assistance of protein regulatory factors.
DOI: https://doi.org/10.7554/eLife.33908.001

*For correspondence:
ronald.breaker@yale.edu

**Competing interests:** The authors declare that no competing interests exist.

## Introduction

A variety of riboswitch classes regulate gene expression in response to the binding of specific metabolite or inorganic ion ligands (*Roth and Breaker, 2009*; *Serganov and Nudler, 2013*; *Sherwood and Henkin, 2016*). Previously, 25 riboswitch classes were known that selectively respond to ligands that are chemical derivatives of RNA monomers or their precursors (*McCown et al., 2017*). This strong bias in favor of RNA-like ligands strengthens the hypothesis (*Nelson and Breaker, 2017*; *Breaker, 2010*) that many of the widely-distributed riboswitch classes are direct descendants from the RNA World – a time before genetically-encoded protein biosynthesis (*Gilbert, 1986*; *Benner et al., 1989*). If this speculation is generally correct, we can expect that some of the most abundant riboswitch classes remaining to be discovered also will regulate gene expression in response to the binding of other ancient RNA derivatives (*Breaker, 2011*).

With these considerations in mind, we evaluated the possible functions of variants of recently validated riboswitches for guanidine (*Nelson et al., 2017*). Collectively, guanidine-I riboswitches along with differently-functioning variants were originally called the *ykkC* orphan riboswitch candidate because no ligand had been identified (*Barrick et al., 2004*; *Meyer et al., 2011*). After removing guanidine-I riboswitches (subtype 1), which constituted ~70% of the original *ykkC* motif RNAs, we further separated the remaining collection of *ykkC* RNA variants (subtype 2). Upon further investigation of the RNA sequences, the phylogeny of organisms containing these RNAs, and the genes associated with each instance of the RNA, we concluded that *ykkC* subtype 2 RNAs were populated by multiple riboswitch classes that sensed distinct ligands.

**eLife digest** Life on Earth is magnificently complex. The instructions to make this complexity are stored within the genes of living cells. For many genes, this process involves two steps. First, the cells copy, or transcribe, the information in the gene – which is part of a DNA molecule – into a more portable but shorter-lived molecule called RNA. The RNA can then be transported to other parts of a cell where its information is decoded to make a protein. Proteins are like molecular machines and making proteins is how many genes contribute to keeping life going.

RNA molecules can fold into intricate shapes including structures called riboswitches. These are parts of an RNA molecule that can control how and when the RNA is used to make proteins. Riboswitches detect the presence of other chemicals inside a cell and respond by changing their shape. Binding to the right molecule will move a riboswitch from an "off" to an "on" position, or vice versa. Riboswitches can help to ensure that some proteins are only made when they are needed. For example, a protein that helps to make a certain chemical is only produced when that chemical is in short supply.

Sherlock et al. have now studied riboswitches in a species of bacteria called *Facklamia ignava*. The investigation revealed a new riboswitch structure that detects phosphoribosyl pyrophosphate, or PRPP, which is a building block for RNA and, eventually, DNA molecules. Further examination revealed that these riboswitches are often linked to others that detect guanine molecules – another component of RNA and DNA. This double riboswitch is similar to a logic gate in computing, activating or shutting down protein production in response to changes in the concentrations of both PRPP and guanine.

The findings of Sherlock et al. reveal a system where RNA controls its own production in response to the availability of two of its critical building blocks. This entirely RNA-dependent system adds weight to the so-called 'RNA world' hypothesis which suggests that the earliest forms of life on Earth depended solely on RNA. Molecules of RNA are able to store data like DNA while also behaving as molecular machines like proteins. This potentially helps to explain how the first life could have arisen from simple, non-living chemicals. Riboswitch systems like this are likely to be relics of this ancient time. Also, because they are involved in fundamental processes in living bacteria, riboswitch systems also provide potentially interesting targets for new antibiotics.

DOI: https://doi.org/10.7554/eLife.33908.002

Subtype 2 representatives were sorted according to the gene located immediately downstream from the *ykkC* RNA motif to create revised consensus models for the RNAs within each gene association group. In some cases, additional conserved nucleotides were observed immediately 5′ and 3′ of the three hairpin structures (called P1, P2 and P3) that are characteristic of *ykkC* motif RNAs. This region located downstream of the guanidine binding site of subtype 1 RNAs were found to be unnecessary for guanidine-I riboswitches. In total, these distinct representatives were sorted into four candidate riboswitch classes called subtypes 2a through 2d.

All subtype 2 RNAs appear to be extremely similar because they share many of the same highly-conserved nucleotides and secondary structure features. We were fortunate to have high-resolution crystal structures (*Reiss et al., 2017*; *Battaglia et al., 2017*) of the guanidine-I riboswitch (subtype 1, *Figure 1A*) in hand when evaluating each of these variant subtypes. Nearly all highly-conserved nucleotides shared by *ykkC* RNAs are involved in forming tertiary contacts within the larger folded structure that positions other nucleotides to form the ligand-binding pocket. However, nucleotides that change identity and are characteristic of each subtype are almost exclusively found in close proximity to the known binding pocket of guanidine-I riboswitches (indicated by black asterisks in *Figure 1A*). For subtypes 2a, 2b, and 2c, we speculated that the additional stretches of conserved nucleotides preceding the P1 stem and following the P3 stem likely play a role in ligand recognition, or perhaps provide structural support for an expanded binding pocket. Unfortunately, because these nucleotides are not conserved in guanidine-I riboswitches, we do not currently have a good understanding of their 3D positioning relative to the rest of the aptamer.

It quickly became clear that sequences associated with branched-chain amino acid metabolism appeared highly similar to those associated with glutamate synthases and *natA* transporters, and so

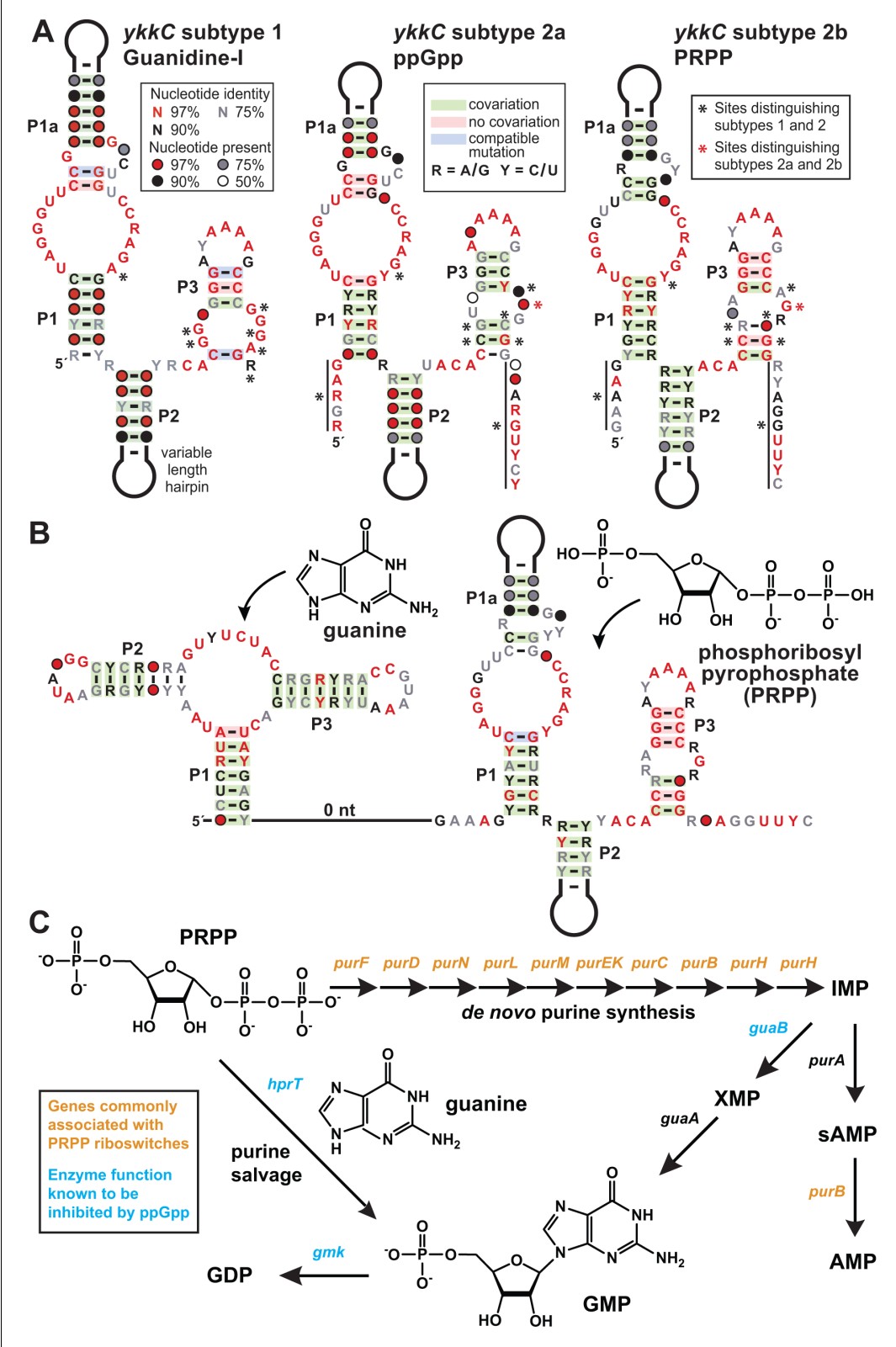

**Figure 1.** PRPP riboswitches and their regulatory network. (**A**) Comparison of the consensus sequence and secondary structure models for the guanidine-I riboswitch aptamer (subtype 1, left) and the aptamers for ppGpp (subtype 2a, middle; ME Sherlock, N Sudarsan, RR Breaker, manuscript submitted), and PRPP (subtype 2b, right). For sequence alignments of the 257 unique PRPP riboswitch aptamers in Stockholm format, see *Supplementary file 2*. (**B**) Consensus sequence and secondary structure model for the 127 unique examples of adjacent guanidine-PRPP aptamers that

*Figure 1 continued on next page*

*Figure 1 continued*

form the corresponding tandem riboswitch architecture. For sequence alignments of the tandem guanine- PRPP riboswitch aptamers in Stockholm format, see *Supplementary file 3*. (C) Only genes in the purine biosynthetic pathway that are necessary for the production of adenosine nucleotides are commonly associated with PRPP riboswitches.

DOI: https://doi.org/10.7554/eLife.33908.003

these were collectively termed *ykkC* subtype 2a (*Figure 1A*). Shortly thereafter, these were experimentally verified as a riboswitch class that responds to the bacterial alarmone ppGpp (ME Sherlock, N Sudarsan, RR Breaker, manuscript submitted). Herein, we describe the function of *ykkC* subtype 2b RNAs (*Figure 1A,B*), whereas the ligands for subtype 2c and 2d RNAs have not yet been identified (see discussion).

Three major clues were exploited to identify phosphoribosyl pyrophosphate (PRPP; 5-phospho-α-D-ribose 1-diphosphate) (*Figure 1B*) as the ligand for riboswitches represented by *ykkC* subtype 2b RNAs. First, representatives of this riboswitch class commonly associate with genes for de novo purine biosynthesis (*Figure 1C*). In addition, cytosine deaminase (*codA*), a gene involved in pyrimidine salvage that is known to be repressed by the purine guanine (*Kilstrup et al., 1989*), is also commonly located downstream of subtype 2b RNAs. These associated genes, which are either related to the production of purines or are regulated by a purine, suggested to us that the ligand would most likely be a compound related to purine metabolism.

Second, there are 257 examples of *ykkC* subtype 2b RNAs distributed throughout bacteria of the phylum Firmicutes. Of these, 127 examples occur in a tandem arrangement (*Sudarsan et al., 2006*) with a guanine aptamer (*Mandal et al., 2003*) located directly upstream (*Figure 1B*). Such tandem arrangements are likely to function in concert as two-input molecular logic gates (; *Stoddard and Batey, 2006*; *Lee et al., 2010*). Guanine riboswitches typically turn off expression of downstream purine biosynthesis genes when sufficient guanine is present (*Mandal et al., 2003*), while representatives of *ykkC* subtype 2b riboswitches were predicted to turn on expression of downstream genes. The physical arrangement of each aptamer in the tandem arrangement suggests that they share a single expression platform. Guanine binding to its aptamer likely would promote the formation of an adjoining intrinsic transcription terminator stem, which would halt transcription. In opposition, binding of ligand to the adjacent subtype 2b RNA would preclude terminator stem formation and thereby override the typical function of the guanine riboswitch. If this speculation is correct, then the ligand for subtype 2b RNAs should trigger cells to make more purines even when one of the major purine compounds, guanine, is plentiful.

Third, we noticed that the consensus sequence and structural models for ppGpp riboswitch aptamers (subtype 2a) and subtype 2b RNAs are remarkably alike, including many similarities at the ligand binding site, suggesting that their ligands might also share some chemical features. Particularly notable was the observation that a long-range C-G base-pair present in guanidine-I riboswitches (*Reiss et al., 2017*; *Battaglia et al., 2017*) is absent in ppGpp riboswitches, but appears to again be present in subtype 2b RNAs. We speculated that the structural role of the G nucleotide missing in ppGpp aptamers might be performed by the guanine base of the ligand. However, because the highly-similar subtype 2b RNA retains this G nucleotide (indicated by a red asterisk in *Figure 1A*) in its aptamer, its ligand might resemble an apurinic version of ppGpp.

## Results

### Variant riboswitch aptamers selectively bind the nucleotide precursor phosphoribosyl pyrophosphate (PRPP)

To evaluate PRPP binding and riboswitch aptamer function, we examined a 106 nucleotide singlet aptamer arbitrarily derived from the *purC* gene of the bacterium *Facklamia ignava* (*Figure 2A*) by using in-line probing (*Soukup and Breaker, 1999*; *Regulski and Breaker, 2008*). This method takes advantage of the inherent chemical instability of phosphodiester linkages in unstructured regions of an RNA and the folding differences in aptamers brought about by ligand binding to provide details on riboswitch function.

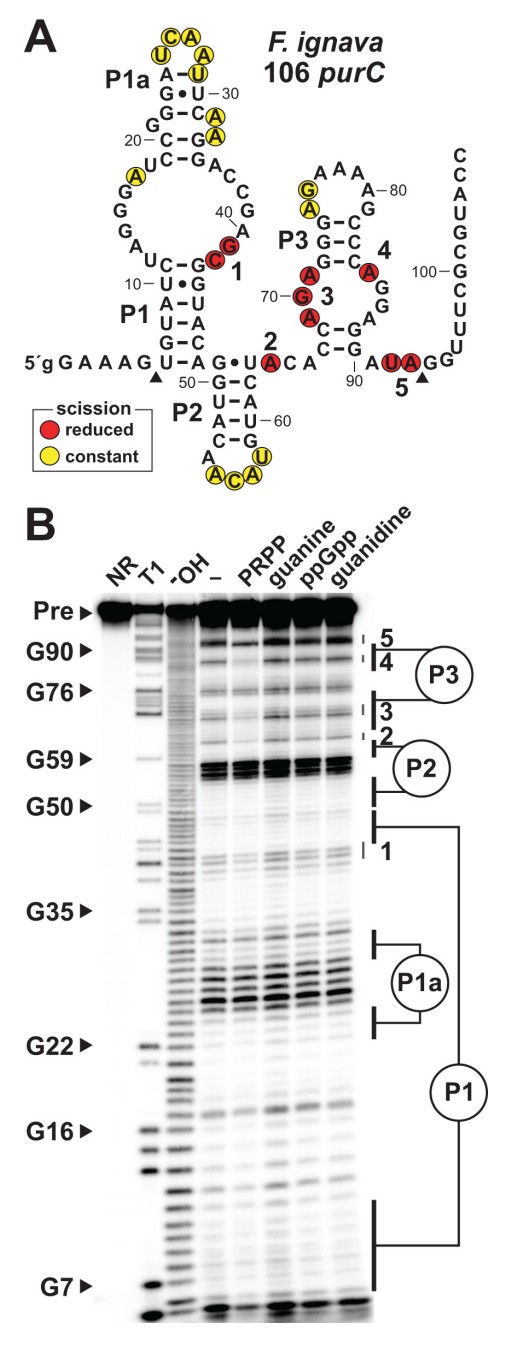

**Figure 2.** Ligand binding by a natural singlet PRPP riboswitch. (**A**) Sequence and secondary structure of the 106 nucleotide RNA derived from the *purC* gene of *F. ignava*. Data collected in B were used to determine regions of constant or reduced scission upon the addition of PRPP to in-line probing assays. Lowercase g indicates a guanosine nucleotide added to enable efficient transcription by T7 RNA polymerase. Arrowheads demark the approximate range of nucleotides with in-line probing data available as depicted in B. (**B**) Polyacrylamide gel electrophoresis (PAGE) analysis of the spontaneous RNA cleavage
*Figure 2 continued on next page*

The 106 *purC* RNA construct exhibits structural modulation at highly conserved nucleotide positions upon the addition of PRPP (*Figure 2B*). In contrast, no structural modulation of the RNA is observed when the known ligands for the other *ykkC* subtypes (guanidine and ppGpp) or for the guanine riboswitch (guanine) are added. Importantly, the structural changes selectively brought about by PRPP largely occur in regions equivalent to those previously implicated in ligand binding by guanidine-I (*Nelson et al., 2017*; *Reiss et al., 2017*; *Battaglia et al., 2017*) and ppGpp riboswitch classes. This observation strongly suggests that these binding site nucleotides have changed to alter ligand specificity, and that PRPP is recognized by this converted binding pocket.

## PRPP triggers riboswitch-mediated regulation of transcription termination

Due to the well-characterized instability of the PRPP molecule (*Dennis et al., 2000*; *Meola et al., 2003*), especially under alkaline conditions and high $Mg^{2+}$, PRPP extensively degrades into multiple breakdown products over the course of our binding experiments. As a result, in-line probing assays will not yield accurate dissociation constants. Therefore, to evaluate the effects of PRPP on riboswitch function, we performed single-turnover in vitro transcription termination experiments (*Wickiser et al., 2005a*), which involve briefer incubations under milder conditions.

The P3 helix of the PRPP aptamer (*Figure 1A*) is predicted to act as an antiterminator stem, which should increase the number of full-length mRNA transcripts when PRPP is bound. Transcription of a singlet PRPP riboswitch derived from the *purB* gene of *Heliobacterium modesticaldum* (*Figure 3A*) produces a greater yield of full-length RNA when PRPP is added to the reaction (*Figure 3B*). The concentration of PRPP necessary to achieve half maximal termination ($T_{50}$) is less than 100 µM (*Figure 3C* and *Figure 3—figure supplement 1*). In contrast, an RNA construct (M1) with disruptive mutations within the highly-conserved aptamer sequence yields the same fraction of terminated and full-length transcripts regardless of PRPP concentration (*Figure 3B*).

Importantly, $T_{50}$ values should not be considered equivalent to $K_D$ values. Riboswitches evaluated by transcription termination assays operate under kinetic limitations, whereas $K_D$ values established by in-line probing can reach thermodynamic equilibrium (*Wickiser et al., 2005a*; *Wickiser et al., 2005b*; *Gilbert et al., 2006*;

*Figure 2 continued*

products generated during in-line probing of 5'-$^{32}$P-labeled 106 *purC* RNA. NR, T1 and $^-$OH represent RNA undergoing no reaction, partial digest with RNase T1 (cleaves after guanosine nucleotides), or partial digest under alkaline conditions (cleaves after every nucleotide), respectively. Bands corresponding to selected G residues are annotated. In-line probing experiments contained either no ligand (–) or 1 mM of the compound indicated except guanine which was tested at 10 μM. Annotations 1 through 5identify prominent regions of the RNA that undergo structural stabilization in a PRPP-dependent manner. Data shown are representative of multiple experiments.

DOI: https://doi.org/10.7554/eLife.33908.004

*Rieder et al., 2007*). Thus, the $K_D$ for PRPP binding to the *H. modesticaldum* aptamer is likely to be far better than the $T_{50}$ values we measured for the *H. modesticaldum purB* construct, which range between 40 and 90 μM (*Figure 3C* and *Figure 3—figure supplement 1*).

## In vivo regulation of gene expression by a PRPP riboswitch

The same singlet PRPP riboswitch sequence from *H. modesticaldum* (*Figure 3A*) was fused to *lacZ* in wild-type (WT) and Δ*purF Bacillus subtilis* cells. Deletion of the *purF* gene, which encodes the enzyme that catalyzes the first committed step of de novo purine biosynthesis (*Figure 1C*), causes PRPP to accumulate when cells are starved for purines. Accordingly, the *lacZ* gene is expressed (blue) in glucose minimal medium in the Δ*purF* strain, but not (clear) in the parental *B. subtilis* strain with a functional *purF* gene (*Figure 3D*).

Notably, the cells range from completely clearto a robust blue under these two conditions, indicating that gene expression is fully suppressed when no ligand is present. Thus, the riboswitch appears to sample a much greater dynamic range for gene expression in vivo than is observed with in vitro transcription termination assays. A riboswitch-reporter fusion carrying a mutation (M2) in the terminator stem expresses *lacZ* under both conditions. These results demonstrate that PRPP accumulation in cells induce gene expression, which parallels our findings in vitro that PRPP binding by *ykkC* subtype 2b RNAs promotes transcription elongation (*Figure 3B* and *Figure 3C*).

## Natural examples of tandem riboswitch aptamers for guanine and PRPP bind these ligands to induce mutually-exclusive structures

We next sought to characterize the tandem arrangement of the guanine and PRPP aptamers by in-line probing analysis of the 208 nucleotide RNA derived from the *codA* gene of *Bacillus megaterium* (*Figure 4A*). Each aptamer appears to independently fold to form its characteristic structure and each exhibits typical structural modulation (*Figure 4B*) upon addition of its cognate ligand as previously observed for singlet guanine (*Mandal et al., 2003*) or PRPP (*Figure 2B*) aptamers.

Only the aptamer regions of the RNA were examined by these in-line probing experiments, but the logic gate function of the tandem riboswitch is accomplished through interactions between the two aptamers and the adjoining expression platform. Consequently, terminator and antiterminator stems are stabilized or destabilized depending on the relative concentrations of both guanine and PRPP.

The precise mechanism by which the guanine and PRPP aptamers work in opposition to control the formation of the terminator stem has yet to be fully experimentally evaluated. However, bioinformatic and in-line probing data already might provide clues regarding the structural interplay between the two aptamers. First, guanine aptamers are always located almost immediately adjacent to the conserved nucleotides at the 5′ terminus of the PRPP aptamer in tandem arrangements (*Figure 1B*). Specifically, there are typically only 5 to 7 nucleotides separating the P1 stems of the guanine and PRPP aptamers, suggesting this juxtaposition is important for logic gate function. Second, the addition of PRPP to the tandem construct causes a reduction in spontaneous RNA cleavage within the conserved nucleotides at the 5′ terminus of the PRPP aptamer (*Figure 4B*, region 5), whereas guanine addition causes this same region to experience a dramatic increase in scission. These results suggest that guanine binding to its aptamer destabilizes an important structural feature necessary for ligand binding by the PRPP aptamer. Thus, each aptamer might interfere with its neighboring aptamer upon ligand binding, perhaps by causing steric clashes due to their proximity. Additional experiments will need to be conducted to more fully evaluate this mechanistic hypothesis.

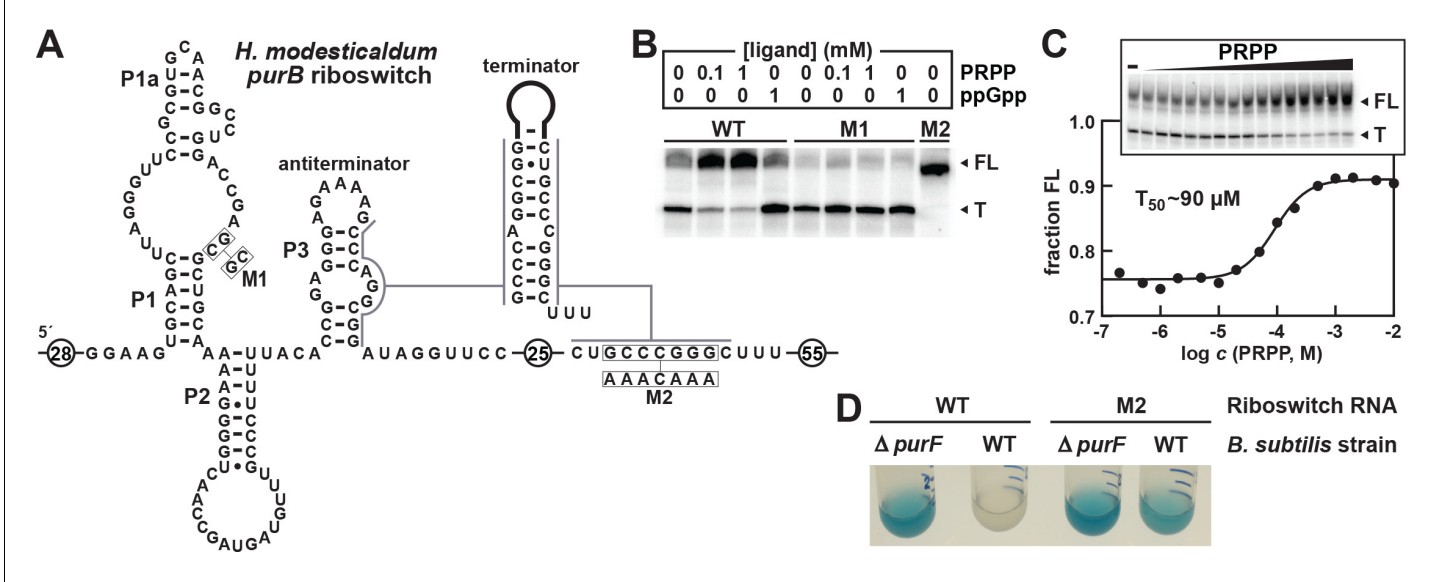

**Figure 3.** Transcription regulation and reporter gene expression by a natural singlet PRPP riboswitch. (**A**) Sequence and secondary structure of the natural PRPP singlet riboswitch derived from the *purB* gene from *H. modesticaldum*. An alternative RNA structure is depicted in which the terminator stem forms followed by a U-rich tract, which only forms in the absence of PRPP. (**B**) PAGE analysis of a single-round transcription termination assay of WT and mutant *H. modesticaldum purB* riboswitches at the indicated ligand concentration. FL and T denote full length and terminated transcripts, respectively. Values for the fraction of full-length transcripts relative to the total transcription yield is listed for each reaction. (**C**) Plot of the fraction of full length WT *H. modesticaldum purB* RNA riboswitch transcripts contributing to the total number of transcripts (FL plus T) as a function of the logarithm (base 10) of the molar PRPP concentration. The concentration of PRPP required to cause half-maximal termination efficiency ($T_{50}$) was determined by a sigmoidal curve fit (see METHODS). Inset: PAGE analysis of single round transcription termination assays of the WT *H. modesticaldum purB* RNA with either no ligand (–) or PRPP ranging from 200 nM to 10 mM. Data for replicates of this transcription termination assay are presented in *Figure 3—figure supplement 1*. (**D**) Reporter gene expression of WT and Δ*purF* [BKE06490 (depicted) and BKK06490 (not shown)] *B. subtilis* cells containing wild-type (WT) or mutant (M2) *H. modesticaldum purB* riboswitch-*lacZ* reporter fusion constructs as described in A. Cells were grown in glucose minimal medium (GMM) containing 50 µg mL⁻1 X-gal.

DOI: https://doi.org/10.7554/eLife.33908.005

The following figure supplement is available for figure 3:

**Figure supplement 1.** Replicate in vitro transcription termination assays with the *H.*

DOI: https://doi.org/10.7554/eLife.33908.006

## Boolean logic function by a riboswitch carrying tandem guanine and PRPP aptamers

To evaluate the proposed two-input logic gate function, the same guanine-PRPP tandem aptamer system including its associated expression platform from *B. megaterium* (*Figure 5A*) was subjected to in vitro transcription termination assays. Consistent with results from the singlet aptamer, the addition of PRPP increases the fraction of RNA transcripts that pass beyond the intrinsic terminator stem and reach full length (*Figure 5B*). Also as predicted, the addition of guanine causes an increase in the fraction of terminated transcripts. When both guanine and PRPP are at high levels, the predicted genetic override occurs, and the fraction of terminated products approaches zero.

Separate and simultaneous mutations to each aptamer (constructs M3 through M5) confirm the OFF and ON function of the guanine and PRPP aptamers, respectively. Specifically, the mutations in M3 alter two highly-conserved nucleotides in the core of the guanine aptamer that are known to be essential for forming its ligand-binding core (*Batey et al., 2004*; *Serganov et al., 2004*). As expected, the addition of guanine does not increase transcription termination. Similarly, the mutations in M4 alter two highly-conserved nucleotides in PRPP aptamer, which we expect to disrupt PRPP binding. Again as expected, the amount of transcription termination product increases, and PRPP addition has no effect on the distribution of transcription products. Construct M5 combines the M3 and M4 mutations to generate an RNA that should fail to bind either ligand. Indeed, the M5 construct appears to be unaffected by the addition of either guanine or PRPP (*Figure 5B*). Similar

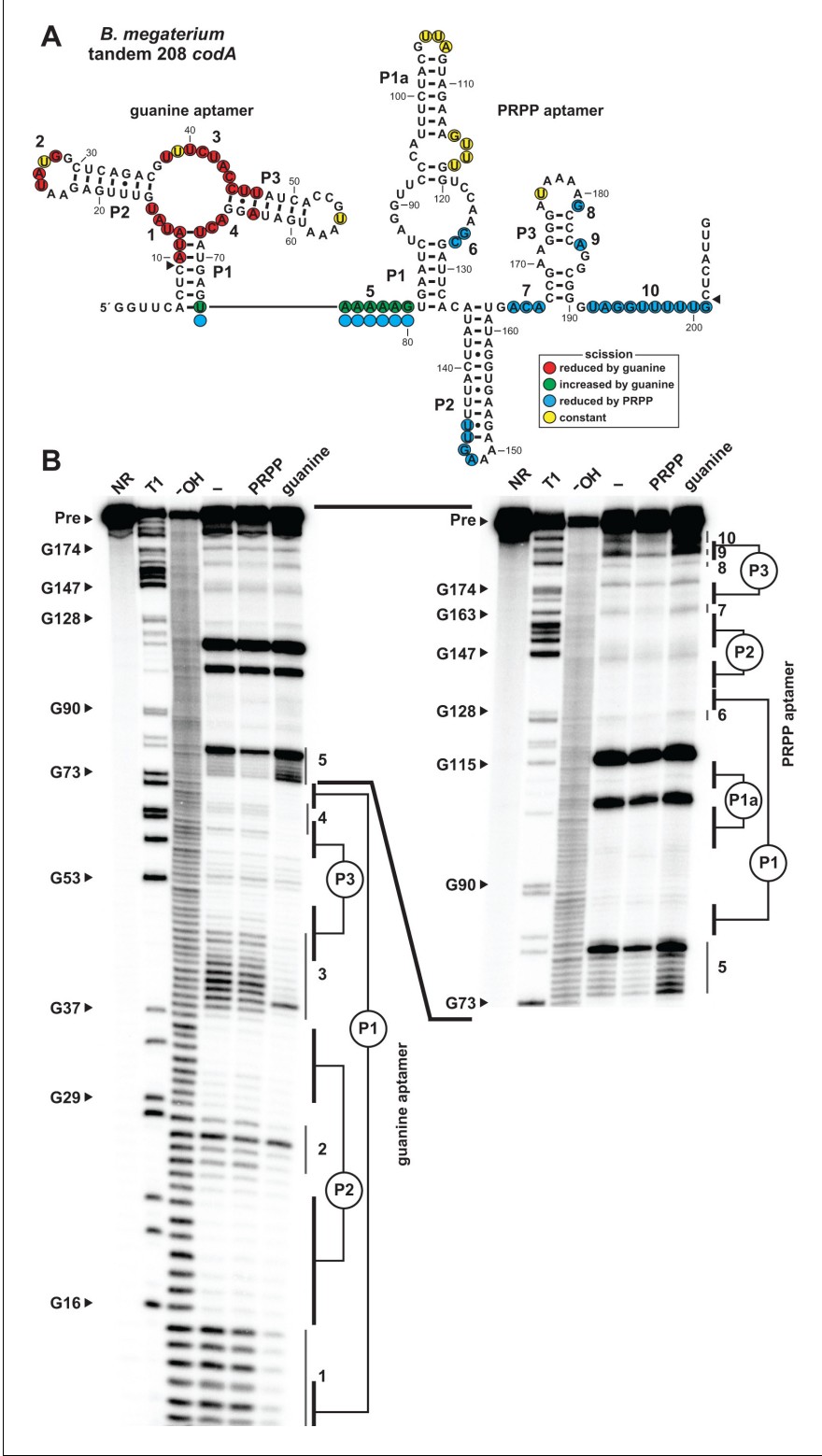

**Figure 4.** Ligand binding by tandem guanine and PRPP aptamers. (**A**) Sequence and secondary structure of the 208 nucleotide RNA derived from the *codA* gene of *B. megaterium*. Data collected in B were used to determine regions of constant, increased, or reduced scission upon the addition of guanine or PRPP to in-line probing assays. Additional annotations are as described for *Figure 4A*. Note that the precise locations of regions experiencing strand scission are approximated after nucleotide 74, as guided by the related in-line probing data generated for *Figure 4 continued on next page*

*Figure 4 continued*

the singlet PRPP aptamer depicted in *Figure 2*. (B) PAGE analysis of the spontaneous RNA cleavage products generated during in-line probing of 5′-$^{32}$P-labeled 208 *codA* RNA. Annotations are described in the legend of *Figure 2B*. In-line probing experiments contained either no ligand (–), 1 mM PRPP or 10 μM guanine. Annotations 1 through 10 identify prominent regions of the RNA that undergo structural stabilization from either guanine or PRPP. Both gel images contain RNA from the same in-line probing reactions, but the gel in the image on the right was run for a longer time to increase resolution of the 3′ region of the RNA, which contains the regions of the PRPP aptamer involved in ligand recognition.

DOI: https://doi.org/10.7554/eLife.33908.007

results are observed in vivo when the WT and M3 through M5 tandem guanine-PRPP riboswitch constructs from *B. megaterium* were fused to the *lacZ* reporter gene and transformed into WT *B. subtilis* cells (*Figure 5C*).

The versatility of this tandem riboswitch system is best demonstrated by varying the concentrations of both guanine and PRPP in a series of in vitro transcription termination assays. The data from these reactions yields a 3D landscape demonstrating again that the relative concentrations of guanine and PRPP dictate the fraction of RNA transcripts that reach full length (*Figure 5D*). Thus, the tandem riboswitch arrangement permits cells to tune the expression levels of associated genes based on the relative concentrations of guanine and PRPP, such that high PRPP levels promote purine production despite the abundance of guanine, a prominent purine.

At the single molecule level, each riboswitch aptamer in this tandem construct favors either transcription termination or elongation based on the binding of its cognate ligand. In this regard, each RNA molecule carrying this tandem aptamer arrangement functions as a Boolean IMPLY logic gate. The truth table for an IMPLY logic gate (*Figure 6A*) indicates that gene expression should be on when neither ligand is present, and when PRPP is bound by the RNA – even when guanine is available. Finally, gene expression should be off when guanine is bound by the RNA, which should occur when guanine is abundant while PRPP is not. This regulatory pattern could be accomplished by the tandem RNA arrangement through formation of an antiterminator stem by alternative base pairing interactions between the two aptamers, similar to those observed for an allosterically-regulated tandem riboswitch-ribozyme described previously (*Lee et al., 2010*; *Chen et al., 2011*). Alternatively, as noted earlier, the aptamers might compete sterically due to the proximity of their conserved sequences and structures. Regardless of the precise mechanism, we hypothesize that an unoccupied PRPP aptamer permits formation of the terminator stem, whereas an occupied guanine aptamer destabilizes the PRPP aptamer. Additional detailed kinetic and structural studies will be necessary to reveal the precise molecular interactions that yield IMPLY gene control logic.

When transcription termination or gene expression is observed as the average output from a population of RNA constructs, the resulting data do not represent a perfect Boolean function because the output yields intermediate values between zero and maximal values. Whereas each of the individual RNAs operates as a binary logic gate that is either on or off, a population of these RNAs exceeds the function of an individual Boolean logic gate. Together, the output of the tandem riboswitch population functions more like a molecular rheostat to tune gene expression based on relative ligand concentrations, rather than as an all-or-nothing binary switch. Regardless, when ligand concentrations are at their extreme values, the output of the tandem system (*Figure 5D*) approaches the binary logic of an IMPLY gate.

## The biological utility of the opposing guanine and PRPP signals evaluated by tandem riboswitch aptamers

A situation in cells in which both guanine and PRPP are scarce would indicate an extremely starved state. Ordinarily, one might expect guanine and PRPP levels to be negatively correlated, such that an excess of guanine would deplete PRPP as a result of the synthesis of 5′-guanosine monophosphate, and vice versa, either through de novo purine biosynthesis or the purine salvage pathway (*Figure 1B*). If true, then why is it necessary to form this complex tandem arrangement to override the gene control function of a guanine riboswitch? This apparent regulatory paradox is addressed in more detail below.

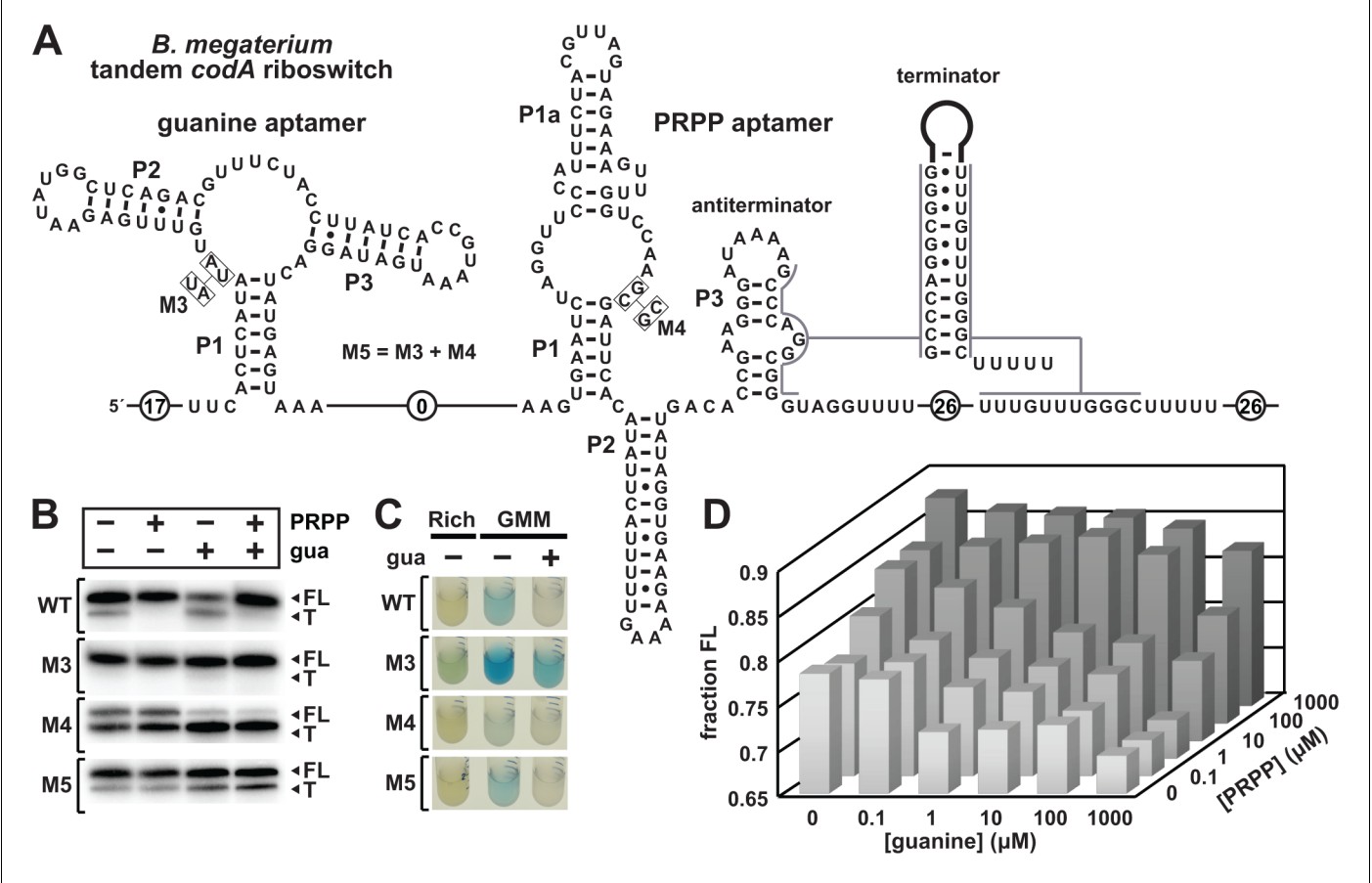

**Figure 5.** Guanine and PRPP competitively regulate transcript termination via a riboswitch integrating two aptamers. (A) Sequence and secondary structure of the tandem guanine-PRPP riboswitch derived from the *codA* gene of *B. megaterium*. An alternative RNA structure is depicted to include the terminator stem followed by a U-rich tract, which forms based on the relative abundance of each ligand. (B) PAGE analyses of single-round transcription termination assays of WT and various mutant *B. megaterium codA* tandem riboswitches in the presence or absence of the two ligands. FL and T denote full length and terminated transcripts, respectively. (C) Reporter gene expression of *B. subtilis* containing wild-type (WT) and mutant (M3 through M5) *B. megaterium* tandem riboswitch-*lacZ* reporter fusion constructs as described in (A). Cells were grown in rich (lysogeny broth) medium or glucose minimal medium (GMM) containing either no (–) or 10 μM additional guanine (gua) and 50 μg mL⁻1 X-gal. (D) Plot of the fraction of full length *B. megaterium codA* tandem riboswitch transcripts contributing to the total number of transcripts as a function of both guanine and PRPP concentrations. Data shown are the average of three experiments (see *Figure 5—figure supplement 1*).

DOI: https://doi.org/10.7554/eLife.33908.008

The following figure supplement is available for figure 5:

**Figure supplement 1.** Comprehensive dataset for the in vitro transcription termination assays with the *B. megaterium codA* riboswitch construct.
DOI: https://doi.org/10.7554/eLife.33908.009

During the stringent response in Firmicutes, nutrient starvation is signaled by the presence of uncharged tRNAs that activate the ppGpp synthase RelA (*Geiger and Wolz, 2014*). ppGpp synthesis causes GTP levels to drop and the transcriptional repressor CodY, which requires GTP and isoleucine to bind its target sites, to release and enable transcription of branched-chain amino acid (BCAA) biosynthesis genes (*Sonenshein, 2005*). The resulting increased ppGpp concentration promotes binding of this signaling molecule as an inhibitor of enzymes such as GuaB, GMK, and HprT (*Figure 1C*) to suppress GTP production (*Kriel et al., 2012*), causing both guanine and PRPP to accumulate.

Under these conditions the guanine-PRPP tandem riboswitch still turns on expression of its downstream genes, despite the mechanisms in place to keep GTP levels low. At first this appears counterintuitive, but the genes associated with the tandem riboswitch system only include those for catalyzing the steps of de novo purine synthesis up to the production of inosine monophosphate

(IMP), the branch point between AMP and GMP production (*Figure 1C*). Thus, the riboswitch promotes AMP synthesis, but will not turn on genes specific for GDP or GTP synthesis that are already being inhibited by ppGpp. ATP is known to be present in elevated quantities during the stringent response because it is used as the initiating nucleotide for transcription of genes under the ppGpp superregulon, whereas rRNA transcripts utilize GTP as the initiating nucleotide to suppress protein production (*Krásný et al., 2008*).

## Expansion of natural Boolean logic devices made of RNA

As noted above, the guanine-PRPP tandem riboswitch arrangement at its functional limits most closely approximates an IMPLY logic gate (*Figure 6A*). This all-RNA system joins the NOR gate formed by tandem *S*-adenosylmethionine (SAM) and adenosylcobalamin (AdoCbl or coenzyme B$_{12}$) riboswitches (*Sudarsan et al., 2006*) (*Figure 6B*) and the AND gate formed by tandem T box and ppGpp riboswitch RNAs (*Figure 6C*) as sophisticated regulatory RNAs that approximate Boolean logic functions.

Similarly, the tandem riboswitch-ribozyme arrangement formed by a c-di-GMP-II riboswitch and a group I self-splicing ribozyme senses both c-di-GMP and guanosine (or any of its 5′-phosphorylated derivatives) to approximate the function of a Boolean AND gate (*Lee et al., 2010*; *Chen et al., 2011*). In this example, one chemical input is sensed by a conventional riboswitch aptamer, whereas the second chemical input is sensed by the active site of a ribozyme. These more complex riboswitch and ribozyme arrangements are notable because they demonstrate how sensory and regulatory systems comprised entirely of RNA can perform sophisticated decision-making in modern cells.

## Discussion

Most experimentally validated riboswitch classes sense and respond to ligands originating from RNA metabolism pathways, and therefore it seemed reasonable to consider the likelihood that riboswitches for PRPP would exist. PRPP is an essential precursor for the modern biosynthesis of RNA monomers in organisms from all three domains of life, and is likely to have been important for the production of ribonucleotides very early in evolution. If PRPP was used by RNA World organisms to generate the building blocks of this major biopolymer, then RNA aptamers that selectively bound this biosynthetic precursor in ancient times might have persisted to serve as riboswitch aptamers in modern cells.

Intriguingly, we had representatives of PRPP riboswitches on our list of riboswitch candidates over a decade ago (*Barrick et al., 2004*), but due to their striking similarity to riboswitches for guanidine (*Nelson et al., 2017*) and ppGpp, they were unusually difficult to identify and experimentally validate. Members of other riboswitch classes have previously been found to exhibit altered ligand specificities by accruing key mutations in nucleotides of their binding pockets. For example, guanine riboswitches have undergone one or a few key mutations to change their ligand specificity to adenine (*Mandal and Breaker, 2004*) or 2′-deoxyguanosine (*Kim et al., 2007*; *Weinberg et al., 2017*). Also, subtle binding site variants of FMN riboswitches appear to bind a closely related ligand (https://patentscope.wipo.int/search/en/detail.jsf?docId=WO2011097027&recNum=27&docAn= US2011000204&queryString=(riboswitch)%2520&maxRec=179; *Weinberg et al., 2017*), and riboswitches for molybdenum cofactor (MoCo) are proposed to have modest binding site changes to alter their ligand specificity to bind tungsten cofactor (*Regulski et al., 2008*). In nearly all these previous examples, there are relatively small changes in the chemical structure of the ligand that need to be accommodated by the riboswitch binding pocket changes.

For the subtypes of *ykkC* motif RNAs whose ligands have been identified, the chemical differences in the ligands can be substantial. The ppGpp (subtype 2a) and PRPP (subtype 2b) riboswitch ligands differ by the presence or absence of a guanine nucleobase and by the location and number of phosphate moieties. Guanidinium, the ligand for guanidine-I (subtype 1) riboswitches is entirely distinct from PRPP, and this moiety forms only a portion of the purine ring of ppGpp. The ligands sensed by subtypes 2 c and 2d, which are associated with genes encoding nucleoside diphosphate linked to X (NUDIX) hydrolases of unknown specificity and a transporter of unknown function associated with phosphonate utilization (phn_DUF6), respectively, still remain to be discovered.

The consensus models of *ykkC* subtype 2c and 2d RNAs each have more prominent changes compared to subtype 2a versus 2b at nucleotide positions that are known to be involved in ligand

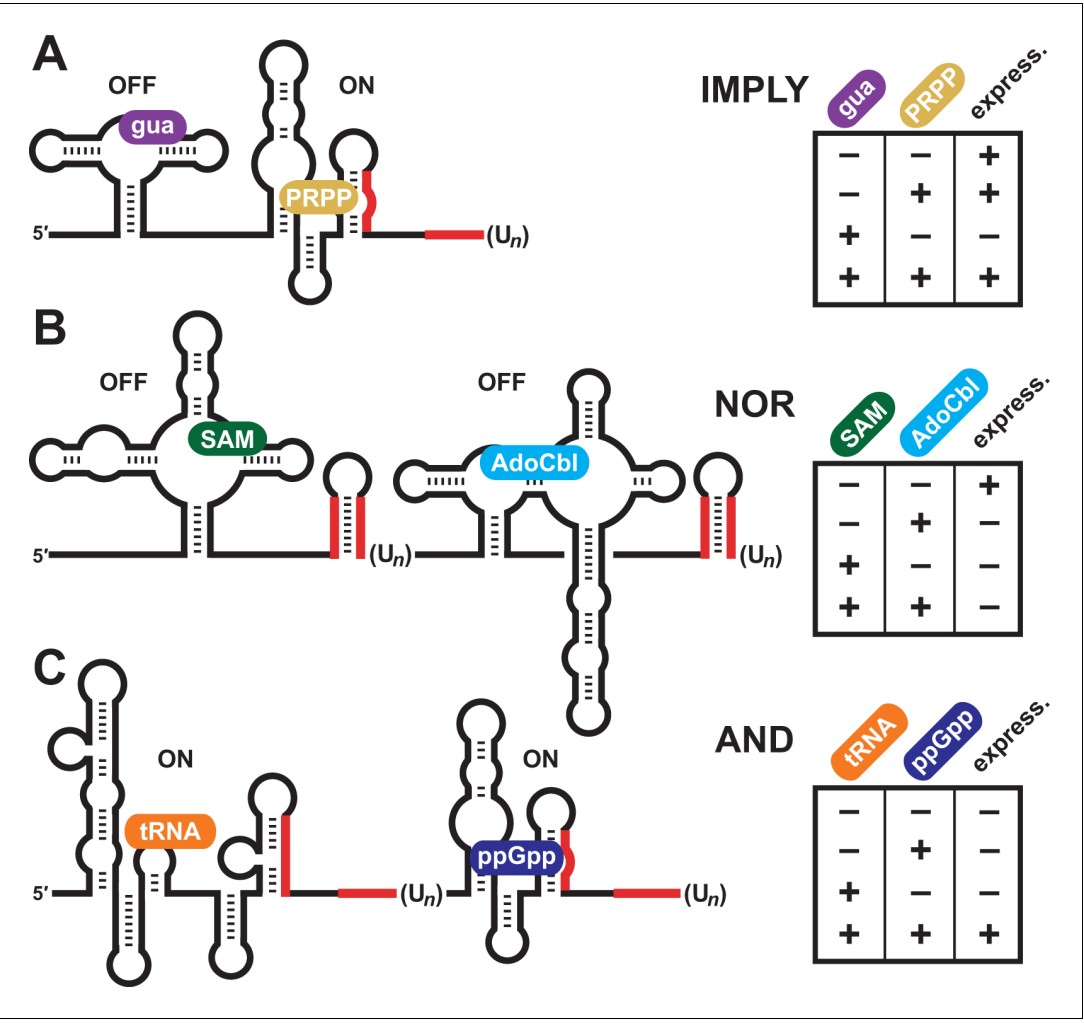

**Figure 6.** Natural examples of tandem riboswitches that approximate Boolean logic functions. (**A**) Tandem guanine and PRPP aptamers form an IMPLY logic gate to determine the state of downstream gene expression (express.). (**B**) Tandem S-adenosylmethionine (SAM) and adenosylcobalamin (AdoCbl) riboswitches form a NOR logic gate. (**C**) Tandem T box RNA and ppGpp riboswitches form an AND logic gate. The T box and ppGpp riboswitch elements can occur in either order.
DOI: https://doi.org/10.7554/eLife.33908.010

recognition (*Nelson et al., 2017*; *Reiss et al., 2017*; *Battaglia et al., 2017*). Because the consensus model for subtype 2c RNAs has the most similarity to those for PRPP and ppGpp riboswitches, and the associated genes encode enzymes with various nucleotide substrates, we speculate that this candidate riboswitch class senses another nucleotide derivative.

The consensus model for subtype 2d RNAs has the most similarity to that for guanidine-I riboswitches and the phn_DUF6 transporter proteins whose expression they control fall into the same drug/metabolite transporter superfamily as guanidine transporters. Therefore, we propose that this unsolved candidate riboswitch class might sense a small, charged, toxic compound akin to guanidine. It will be interesting to compare the chemical structures of all of the *ykkC* family riboswitch ligands once those for the subtype 2c and 2d riboswitch candidates have been validated. Nonetheless, the collection of *ykkC* motif RNAs already demonstrates the ability of RNA to recognize a variety of distinct small molecules, all while utilizing the same general structural scaffold.

Furthermore, the discovery of tandem guanine-PRPP riboswitches serve as additional demonstrations of how complex molecular devices can be formed purely from RNA. We should note here that the in vitro transcription data for the logic gate molecules (*Figure 5B,C*) represent the combined

output of a population of RNA transcripts. Although the transcript length trends are as predicted for an IMPLY gate, these outputs do not match the perfect binary output states of the corresponding electronic logic gates. However, each individual RNA molecule indeed has a binary output just like an individual electronic logic gate. Each tandem riboswitch aptamer transcript is either terminated (OFF) or fully transcribed (ON), but the band intensities from analytical gel electrophoresis report on the results of the population of transcripts.

The outputs of the riboswitch logic gate can be further complicated by at least two issues. First, an individual transcript might fail to function properly, say by misfolding of an aptamer domain, such that it cannot respond to the presence of a ligand. This failure mode might help explain the imperfect yields of terminated and full-length transcripts observed when examining riboswitches by in vitro transcription assays. It seems likely that folding problems of RNAs prepared in vitro might be less prominent when the RNAs are produced by cells. This means that high levels of transcription read-through that is non-responsive to the absence of ligand are not evident in reporter gene assays because the riboswitch aptamers have evolved to fold properly and on a relevant timescale in the cellular milieu.

Second, the presence or absence of the two ligand inputs will change the function of the logic gate system at the individual transcript level, and therefore change the frequency of their choice of these two states rather than change the number of output states possible. Unlike electronic logic gates that use electrical current at only two values (off and on), ligand concentrations for riboswitches can be highly variable. Therefore, given the nature of ligand binding to receptors (each described by an equilibrium constant equation), differences in ligand concentrations will result in differences in the probabilities that the two riboswitch aptamers will be occupied. This occupancy distribution will be reflected in the transcriptional output of the RNA population, but each individual transcript still yields a binary output (terminated or full-length product) that is biased by the concentrations of ligands present.

The IMPLY gate described herein involves a riboswitch RNA with aptamers for the RNA nucleobase guanine and the RNA precursor PRPP. While tandem riboswitch systems have previously been described (*Figure 6*), the guanine-PRPP system is the only example of two tandem aptamers with different ligand specificities that share the same expression platform, allowing for a new type of logic function to be implemented. All of these RNA systems demonstrate complex chemical sensing and regulatory functions, which perhaps reflects the complexity of systems used for metabolite monitoring and biological regulation in ancient RNA World organisms. Furthermore, it seems likely that additional variations of two-input Boolean logic gates will be represented by tandem riboswitches that have yet to be discovered in modern cells.

## Materials and methods

### Chemicals and reagents

Chemicals were purchased from Sigma-Aldrich with the exceptions of phosphoribosyl pyrophosphate (Santa Cruz Biotech) and guanosine 5′,3′-bisdiphosphate (TriLink Biotech). [γ-$^{32}$P]-ATP and [α-$^{32}$P]-UTP were purchased from Perkin Elmer and used within two weeks of receipt. Bulk chemicals were purchased from J.T. Baker and enzymes from New England Biolabs, unless otherwise noted. All solutions were prepared using deionized water (dH$_2$O) and were either autoclaved or filter sterilized (using 0.22 µm filters, Millipore) prior to use. DNA oligonucleotides were purchased from Sigma-Aldrich and Integrated DNA Technologies. A list of oligonucleotides used in this study can be found in *Supplementary file 1*.

### Bioinformatics analyses

Additional examples of *ykkC* motif RNAs were identified using Infernal 1.1 (*Nawrocki and Eddy, 2013*) to search RefSeq version 76 plus additional environmental microbial databases as described previously (*Weinberg et al., 2015*). Iterative searches for new sequences were performed based on the previously published alignment of *ykkC* subtype 2 RNAs (*Nelson et al., 2017*). These sequences were sorted by nucleotide identity at a specific position as previously indicated (*Nelson et al., 2017*) to exclude guanidine-I riboswitches, then further manually sorted by downstream gene association, as described above. This revealed 257 unique examples of subtype 2b RNAs (PRPP riboswitches),

127 of which are found in tandem with a guanine aptamer. The consensus sequence and secondary structure model was constructed using R2R software (*Weinberg and Breaker, 2011*).

## RNA oligonucleotide preparation

Double-stranded DNA (dsDNA) templates for RNA transcription were produced either by extension of overlapping synthetic DNAs (*Supplementary file 1*) using SuperScript II reverse transcriptase (Thermo Fisher Scientific) for the *F. ignava* construct or by PCR from genomic DNA for *B. megaterium* construct. Primers were designed to contain a 5′-terminal T7 RNA polymerase (T7 RNAP) promoter to enable transcription. The desired RNA constructs were prepared from these dsDNAs by in vitro transcription, purified, and subsequently 5′ $^{32}$P-labeled as previously described (*Nelson et al., 2017*; *Sherlock et al., 2017*).

## RNA in-line probing analyses

In-line probing assays (*Soukup and Breaker, 1999*; *Regulski and Breaker, 2008*) were performed precisely as described previously (*Nelson et al., 2017*; *Sherlock et al., 2017*).

## Single-round transcription termination assays

DNA constructs for single-round in vitro transcription were designed to carry the riboswitch of interest spanning from the predicted natural transcription start site to 55 (*H. modesticaldum* construct) or 26 (*B. megaterium* construct) nucleotides following the terminator stem. The promoter from the *B. subtilis lysC* gene, which is compatible with *E. coli* RNA polymerase, was used for both in vitro transcription termination as well as the *lacZ* reporter experiments detailed below. Mutations were incorporated in the region of the *B. megaterium* construct before the guanine aptamer so that it contained no cytidines, while the *H. modesticaldum* construct was naturally cytidine-deficient. Synthetic double-stranded DNA templates were designed, purchased (Integrated DNA Technologies) and subsequently amplified by PCR for the *H. modesticaldum* constructs whereas constructs for the WT and M3 through M5 *B. megaterium* riboswitch were amplified from the plasmids containing those inserts for the reporter gene fusion, as described below.

Approximately 2 pmol of the resulting, purified DNA template was added to a transcription initiation mixture (20 mM Tris-HCl [pH 7.5 at 23°C], 75 mM KCl, 5 mM MgCl$_2$, 1 mM DTT, 10 μg mL$^{-1}$ bovine serum albumin [BSA], 130 μM ApA dinucleotide, 1% glycerol, 0.04 U μL$^{-1}$ *E. coli* RNA polymerase holoenzyme, 2.5 μM GTP, 2.5 μM ATP, and 1 μM UTP). Approximately 1 μCi [α-$^{32}$P]-UTP was added per 8 μL reaction and transcription was allowed to proceed at 37°C for 10 min, leading to formation of a stalled polymerase complex at the first cytidine nucleotide of each transcript. For each 8 μL transcription reaction, 1 μL of 10x elongation buffer (20 mM Tris-HCl [pH 7.5 at 23°C], 75 mM KCl, 5 mM MgCl$_2$, 1 mM DTT, 2 mg mL$^{-1}$ heparin, 1.5 mM ATP, 1.5 mM GTP, 1.5 mM CTP, and 250 μM UTP) as well as 1 μL of a 10x solution of the ligand of interest were added sequentially. Transcription reactions were then incubated at 37°C for an additional 60 min.

The transcription products were subsequently analyzed via denaturing (8 M urea) 10% PAGE and visualized using a phosphorimager (GE Healthcare Life Sciences). Fraction full length values were calculated by varying the ligand concentration in separate reactions and quantifying the changes in band intensity of both full length (FL) and terminated (T) transcription products using the formula (FL)/(FL +T). The T$_{50}$ values were determined by plotting the fraction FL as a function of the logarithm of ligand concentration and using a sigmoidal four parameter logistic fit in GraphPad Prism 7.

## Riboswitch reporter assays

Riboswitch-reporter constructs consisting of the *B. subtilis lysC* gene promoter and either the *H. modesticaldum* or *B. megaterium* riboswitch fused upstream of the *E. coli lacZ* gene were inserted into the vector pDG1661 as described previously (*Sudarsan et al., 2003*; *Nelson et al., 2017*) and integrated into the *amyE* locus of wild-type WT (1A1 strain 168 Δ*trp*) or Δ*purF* strains (BKE06490 [erythromycin resistant] and BKK06490 [kanamycin resistant]) obtained from the Bacillus Genetic Stock Center, The Ohio State University) *B. subtilis* (*Meyer et al., 2011*). The resulting transformed strains were verified to exhibit the expected resistance to chloramphenicol as well as kanamycin for the Δ*purF::kan* strains. Additionally, each strain was confirmed to contain the desired riboswitch reporter sequence by colony PCR amplification followed by sequencing. For example, PCR

amplification using primers flanking the *purF* locus followed by sequencing confirmed either the WT *purF* gene (WT strains) or kanamycin-resistance gene (Δ*purF::kan* strains) as expected. Mutations were incorporated into the *B. megaterium* reporter construct by using Quikchange site-directed mutagenesis (Agilent Technologies).

Reporter experiments were performed by inoculating various *B. subtilis* strains into Lysogeny Broth (LB) with appropriate antibiotics and growing overnight at 37°C. For experiments with the singlet *H. modesticaldum* reporter, the bacteria were diluted into lysogeny broth (LB), grown for an additional 4–6 hr until culture growth was visible, centrifuged, supernatant removed, the resuspended in glucose minimal media (GMM). For experiments with the tandem *B. megaterium* reporter, bacteria were diluted directly into either LB or GMM, as indicated, and grown for 6 hr with or without supplemented guanine. Synthetic DNAs used in cloning are listed in *Supplementary file 1*.

## Acknowledgements

We thank Adam Roth, Ruben Atilho, Kimberly Harris, and other members of the Breaker laboratory for helpful discussions. MES was supported by an NIH Cellular and Molecular Biology Training Grant (T32GM007223). This work was also supported by NIH grants to RRB (GM022778 and DE022340). RRB is also supported by the Howard Hughes Medical Institute.

## Additional information

### Funding

| Funder | Grant reference number | Author |
| --- | --- | --- |
| NIH Office of the Director | GM022778 | Ronald R Breaker |
| Howard Hughes Medical Institute | Investigator | Ronald R Breaker |
| NIH Office of the Director | DE022340 | Ronald R Breaker |
| NIH Office of the Director | T32GM007223 | Madeline E Sherlock |

The funders had no role in study design, data collection and interpretation, or the decision to submit the work for publication.

### Author contributions

Madeline E Sherlock, Conceptualization, Data curation, Formal analysis, Validation, Investigation, Methodology, Writing—original draft, Writing—review and editing; Narasimhan Sudarsan, Conceptualization, Data curation, Formal analysis, Validation, Investigation, Methodology, Writing—review and editing; Shira Stav, Investigation, Writing—review and editing; Ronald R Breaker, Conceptualization, Resources, Supervision, Funding acquisition, Methodology, Writing—original draft, Project administration, Writing—review and editing

### Author ORCIDs

Ronald R Breaker (iD) http://orcid.org/0000-0002-2165-536X

### Decision letter and Author response

Decision letter https://doi.org/10.7554/eLife.33908.016
Author response https://doi.org/10.7554/eLife.33908.017

## Additional files

### Supplementary files

• Supplementary file 1. This file includes a table listing the DNA constructs used in the study.
DOI: https://doi.org/10.7554/eLife.33908.011

• Supplementary file 2. This file presents a sequence alignment in Stockholm format for individual PRPP riboswitch aptamers.

DOI: https://doi.org/10.7554/eLife.33908.012

• Supplementary file 3. This file presents a sequence alignment in Stockholm format for riboswitches that carry tandem guanine and PRPP aptamers.
DOI: https://doi.org/10.7554/eLife.33908.013

• Transparent reporting form
DOI: https://doi.org/10.7554/eLife.33908.014

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
