## [Decision Letter]

Thank you for submitting your article "Tandem riboswitches form a natural Boolean logic gate to control purine metabolism in bacteria" for consideration by *eLife*. Your article has been reviewed by three peer reviewers, and the evaluation has been overseen by Timothy Nilsen as Reviewing Editor and Naama Barkai as the Senior Editor. The following individual involved in review of your submission has agreed to reveal his identity: Ailong Ke (Reviewer #1).

The reviewers have discussed the reviews with one another and the Reviewing Editor has drafted this decision to help you prepare a revised submission.

There was a consensus that this work reports a novel type of RNA-based logic gate that uses a regulatory mechanism distinct from previously described tandem riboswitches. The work was performed rigorously and all the reviewers felt that it was, in principle, suitable for publication in *eLife*. Since all of the points raised by the referees can be addressed by fairly minor textual changes, the entire reviews are included. Please address all comments.

*Reviewer #1:*

The manuscript by Sherlock et al. describes the function and mechanism of a riboswitch class that resembles the guanidine-I riboswitch in structure, but senses a different ligand, PRPP, to regulate the purine synthesis pathway. In the first half of the study, the specificity of these riboswitches for PRPP is proven through in-line probing analysis. This combined with transcription read-through assays and in vivo expression studies provides convincing evidence that these riboswitches only respond to PRPP and not guanidine. In the second part, the authors describe the IMPLY logic gate that PRPP riboswitches form in tandem with guanine riboswitches to regulate purine biosynthesis in starvation conditions. The regulatory properties of these tandem riboswitches in response to different ratios of PRPP and guanine is thoroughly characterized with in vitro and in vivo experiments. This is a novel type of RNA-based logic gate that utilizes a regulatory mechanism distinct from other tandem riboswitches described previously. Therefore, this work has significance to the riboswitch and RNA biology fields.

Overall, the manuscript is polished with only a few areas needing some proof-reading, the conclusions are sound, and the significance qualifies it for the broad readership of *eLife*. There is a major point I would ask the authors to consider.

While the authors convincingly show the regulatory effects of the guanine-PRPP tandem switch, the mechanism by which the two aptamers communicate to alter expression is not given. The authors suggest steric hindrance as a possible mechanism; would extending the linker between the guanine and PRPP domains change their regulatory outcome? Is it possible to make a mutant that retains binding activity in both aptamers, but loses the ability to repress transcription in response to guanine? Furthermore, the authors show in their in-line probing experiments that an AAAAA sequence between the guanine and PRPP domains has altered scission depending on if guanine or PRPP is present. Can the authors suggest a possible role for this sequence?

*Reviewer #2:*

This riboswitch-focused manuscript includes a full range of the methods that are typically used to validate new riboswitch classes; therefore, this represents a thorough validation of a new and important riboswitch class. It is particularly interesting as it discovers PRPP as the ligand for a subclass (ykkC-2b) of a previously identified riboswitch class.

It is a well-written manuscript, with a thorough Discussion and effective figures; I have no criticisms or substantive reservations with the manuscript as written. Therefore, the only real question is whether the manuscript warrants publication in *eLife* as opposed to other more specialty journals. A subset of the reasons that it does are as follows:

- This manuscript is effectively a meaningful "sequel" to a prior publication, which suggested that some bacteria might use tandem riboswitches that respond to combinatorial inputs. The current manuscript reveals how these original tandem riboswitches function, and how they are likely to influence bacterial stress responses.

- The model for the logic behind the tandem riboswitch (subsection “Expansion of natural Boolean logic devices made of RNA”) makes perfect sense. Moreover, it meaningfully contributes to the general understanding of the types of cellular responses that occur during ppGpp signaling.

- While some riboswitch classes have been found to have members that respond to a ligand that is modestly different than is sensed by the overall riboswitch family, the "ykkC" riboswitch class is truly remarkable in this regard. This manuscript demonstrates that the evolutionary malleability of the "ykkC" structural scaffold is incredible, with subclasses that have been shown to respond to guanidinium, ppGpp, and now PRPP, and still with several more uncharacterized classes. This important data reveals that riboswitch classes must be carefully analyzed by bioinformatics techniques in order to try and identify meaningful subclasses. In other words, the emerging "ykkC" story serves as a precautionary reminder that one cannot assume that all members of a riboswitch class behave the exact same, without further bioinformatic or biochemical analysis.

- This manuscript adds substantially to the discussion on how combinations of riboswitches can act as different types of Boolean logic gates. This is important for understanding the complex regulatory signals that underlie genetic regulation of biological functions in bacteria, but also because it helps set the stage for development of synthetic riboswitch RNAs.

The only mark against the manuscript is that it is unable to provide a molecular description of the exact mechanism of allosteric connectivity between the guanine and PRPP aptamers; however, I agree with the authors that this is well beyond the scope of this current manuscript.

*Reviewer #3:*

The study by Sherlock and coauthors presents several findings, which will be of interest to the broad readership of *ELife*. The first major finding is identification of a cognate ligand for a subtype of the orphan ykkC riboswitch. Surprisingly, this ligand, PRPP, is very different from guanidine, a ligand for another subtype of the same riboswitch type. Therefore, the PRPP and guanidine riboswitches are the first example of riboswitches that appear to share the same or very similar structures but recognize ligands with distinct physicochemical properties.

Another major finding is characterization of a tandem riboswitch arrangement involving guanine and PRPP sensors. Integration of the two sensors and a single expression platform results in a complex biological readout that approximates the Boolean IMPLY logic gate. Such tandem riboswitch arrangement adds to the growing collection of complex regulatory outputs from the RNA-only regulatory elements. The study is performed according to the highest standards of the Breaker laboratory and I do not have any major concerns.

I would only recommend the authors to make additional analysis of the linker between guanine and PRPP sensors in the tandem riboswitches. The tandem riboswitch requires overlapping sequences of the sensors to participate in the formation of mutually exclusive ligand-binding structures in order to function as the IMPLY gate. While the majority of the linkers are short (≤5 nt), as reflected in the consensus sequence, a number of them are longer than 5 nt, with ~20% being 7 nt long. The in-line probing experiments were performed on the riboswitch with a 6-nt linker, and while there was no sequence overlap between the sensors, formation of the PRPP sensor affected the 3' nucleotide of the guanine sensor, thereby explaining the IMPLY mechanism. With a 7-mer linker, the IMPLY mechanism may not be efficient and I wonder whether there is a simple explanation for having such outliers. Is the P1 helix of these riboswitches longer than 7 base pairs? If not, these riboswitches could be an interesting system to test in future studies, and the authors may comment on this observation.

---

## [Author Response]

Reviewer #1:[…] Overall, the manuscript is polished with only a few areas needing some proof-reading, the conclusions are sound, and the significance qualifies it for the broad readership of eLife. There is a major point I would ask the authors to consider.While the authors convincingly show the regulatory effects of the guanine-PRPP tandem switch, the mechanism by which the two aptamers communicate to alter expression is not given. The authors suggest steric hindrance as a possible mechanism; would extending the linker between the guanine and PRPP domains change their regulatory outcome? Is it possible to make a mutant that retains binding activity in both aptamers, but loses the ability to repress transcription in response to guanine? Furthermore, the authors show in their in-line probing experiments that an AAAAA sequence between the guanine and PRPP domains has altered scission depending on if guanine or PRPP is present. Can the authors suggest a possible role for this sequence?

This is the major topic for comments from all three reviewers. We had initially judged that a careful assessment of the mechanism of inter-aptamer communication would take considerable biochemical and biophysical research campaign. As a result, in the current manuscript, we only discussed two clues regarding how these aptamers interact (subsection “Natural examples of tandem riboswitch aptamers for guanine and PRPP bind these ligands to induce mutually-exclusive structures”, last paragraph). We felt that a deeper understanding of the mechanism was beyond the scope of our report of the discovery and initial validation of this new riboswitch class. Specifically, we noted that the two aptamers are always located immediately adjacent to each other. In addition, the relative flexibility of the conserved 5’ portion of the PRPP aptamer is dependent on the ligand occupancy of the two aptamers. These two observations allow us to tentatively conclude that this region is likely to be critical for the mutually-exclusive formation of the two aptamers. It now turns out that our speculation is accurate, but we must consider two additional concerns when preparing our revised manuscript as described below:

First, both reviewer 2 and reviewer 3 note that the distance between the two aptamers is not perfectly conserved, and that some natural representatives have one nucleotide more or less compared to the construct we tested. This is true, and therefore we needed to soften our statement in the paragraph summarizing the clues regarding the mechanism of inter-aptamer communication. The relevant sentences now read as follows:

“First, guanine aptamers are always located almost immediately adjacent to the conserved nucleotides at the 5ˊ terminus of the PPRP aptamer in tandem arrangements (Figure 1B). Specifically, there are typically only 5 to 7 nucleotides separating the P1 stems of the guanine and PRPP aptamers, suggesting this juxtaposition is important for logic gate function.”

Second, our Yale colleagues in the laboratory of Scott Strobel have now generated an atomicresolution model of a PRPP aptamer, which demonstrates that the linker sequence indeed is an integral part of the PRPP aptamer structure formed upon binding ligand. Thus, our finding that the guanine riboswitch aptamer (when bound to guanine) destabilizes the structure of this adjacent linker sequence is entirely consistent with a model wherein the two aptamers form in a mutually-exclusive fashion. Because this atomic-resolution structure of the PRPP aptamer was completed independently by the Strobel laboratory, and given that this work has not yet been submitted for publication, we really must restrict our comments on the mechanism. This part of the story should be described by the Strobel laboratory, and therefore we have not added more discussion regarding the mechanism of logic gate function to our revised manuscript.

Reviewer #2:[…] The only mark against the manuscript is that it is unable to provide a molecular description of the exact mechanism of allosteric connectivity between the guanine and PRPP aptamers; however, I agree with the authors that this is well beyond the scope of this current manuscript.

See the response point above.

Reviewer #3:[…] I would only recommend the authors to make additional analysis of the linker between guanine and PRPP sensors in the tandem riboswitches. The tandem riboswitch requires overlapping sequences of the sensors to participate in the formation of mutually exclusive ligand-binding structures in order to function as the IMPLY gate. While the majority of the linkers are short (≤5 nt), as reflected in the consensus sequence, a number of them are longer than 5 nt, with ~20% being 7 nt long. The in-line probing experiments were performed on the riboswitch with a 6-nt linker, and while there was no sequence overlap between the sensors, formation of the PRPP sensor affected the 3' nucleotide of the guanine sensor, thereby explaining the IMPLY mechanism. With a 7-mer linker, the IMPLY mechanism may not be efficient and I wonder whether there is a simple explanation for having such outliers. Is the P1 helix of these riboswitches longer than 7 base pairs? If not, these riboswitches could be an interesting system to test in future studies, and the authors may comment on this observation.

See the response point above.